# Enhancement of the Diversity of Pollinators and Beneficial Insects in Intensively Managed Vineyards

**DOI:** 10.3390/insects12080740

**Published:** 2021-08-18

**Authors:** Francisco Javier Peris-Felipo, Fernando Santa, Oscar Aguado, José Vicente Falcó-Garí, Alicia Iborra, Michael Schade, Claire Brittain, Vasileios Vasileiadis, Luis Miranda-Barroso

**Affiliations:** 1Syngenta Crop Protection, Rosentalstrasse 67, 4058 Basel, Switzerland; fernando.santa@syngenta.com (F.S.); michael.schade@syngenta.com (M.S.); vasileios.vasileiadis@syngenta.com (V.V.); 2Andrena Iniciativas y Estudios Medioambientales S.L., Calle Gabilondo 16bis, 47007 Valladolid, Spain; oscaraguado@lepidopteros.com; 3Laboratory of Entomology and Pest Control, Institute Cavanilles of Biodiversity and Evolutionary Biology, Calle Catedrático José Beltrán 2, Paterna, 46980 Valencia, Spain; j.vicente.falco@uv.es (J.V.F.-G.); lopezmil@alumni.uv.es (A.I.); 4Jeallot’s Hill International Research Centre, Syngenta, Bracknell RG42 6EY, UK; claire.brittain@syngenta.com; 5Agricultura Sostenible Syngenta España, Calle de la Ribera del Loira, 8, 10, 28042 Madrid, Spain; luis.miranda@syngenta.com

**Keywords:** biodiversity, vineyards, agro-ecosystems, sustainability, habitat management, insect conservation, cover plants, natural enemies, pollinators

## Abstract

**Simple Summary:**

The continuous intensification of agricultural production has resulted in higher yields and more yield security. However, these achievements went along with the substitution of heterogeneous agricultural landscapes by homogeneous ones with poor crop diversity, short crop rotations, and thanks to the high efficacy of modern herbicides and also to minimum in-crop diversity. A severe increase in plot size led to the elimination of ecologically valuable structural elements that had provided floral resources and nesting sites. Over the few last decades, several studies have been conducted to try to find solutions against insect decline and to preserve biodiversity. In the present study, the integration of cover plants between the lines of the vineyards to enhance biodiversity is shown. The benefits of the cover plants use are presented based on the results achieved on five intensive wine farms in Spain. Our findings suggest that the use of cover plants provide a wide range of enhancements in the insect community with a significant increase both in the number of species and the number of individuals showing an important influence over time, which would tend to have a significant conservation impact thanks to its effect as a reservoir of species.

**Abstract:**

(1) Modern, intensive agricultural practices have been attributed to the loss of insect biodiversity and abundance in agroecosystems for the last 80 years. The aim of this work is to test whether there are statistically significant differences in insect abundance between different zones and over time on the vineyard field. (2) The study was carried out in five intensive wine farms in Spain over a three-year period (2013–2015). Each field was divided into two zones, one where cover plants were planted, and another remained unchanged (without cover). (3) A clear trend to increase the average number of insect species and individuals throughout the years in all farms was observed. Moreover, the zones with cover plants showed a significant difference with respect to the zones without. (4) The use of permanent cover plants allows creating areas of refuge for the insects favouring their conservation and reducing the agriculture impact in the insect decline.

## 1. Introduction

Since World War II, governments have been supporting farmers to increase their productivity despite the increased costs of production and its environmental impact. Rapid intensification and industrialization of agriculture are often cited as contributing factors in insect declines [1,2,3,4,5].

Land-use changes in the 1950–1970s notably impacted modern intensive agricultural practices, resulting in a substitution of heterogeneous agricultural landscapes by homogeneous ones. A big increase in farm size led to an elimination of edges and other ecologically valuable structural elements that had provided floral resources and nesting sites [4,6,7,8,9,10]. Habitat loss caused vital changes in the natural communities of birds, insects and mammals. In addition, the intensification measures have also led to soil quality losses [3,11,12,13,14].

Pollinating insects have been highlighted as being severely affected by agricultural intensification [15,16,17,18,19,20,21,22]. Within the pollinating insects, wild bees are often the group of insects that suffered the highest decline, reaching up to 50% of populations [19,23,24,25].

Over the last decades, several options were considered to reverse the situation. The use of multifunctional margins to increase the abundance of wildflowers, insects and birds has been highlighted as an important way of promoting nature conservation [3,5,26,27,28,29,30]. However, most of these studies are based on the study of bees or bumble bees [10,22,31,32,33,34], and only a few included all the insect groups [13,35].

In this work, assuming that biodiversity can be measured by the abundance of insects, the differences in several environments are evaluated and verified if their effect is permanent over time. This leads to testing two hypotheses. First, there is a benefit of integrating the cover plants in the inter-row lines of the crop to enhance biodiversity. Second, the use of cover plants improves biodiversity over time. These hypotheses were studied in five intensive wine farms in Spain.

## 2. Materials and Methods

### 2.1. Areas of Study

For this study, five commercial farms were selected in Spain (Table 1). All the established agricultural practices in these farms, such as tilling, fertilization, and phytosanitary treatments, remained unchanged. Management measures were confined to the crop, trying not to interfere with the cover plants.

### 2.2. Mixture Plant Selection

The selection of plant species was based on several fundamental criteria such as the strict use of native species, ensuring a smooth climatic adaptation; being non-weed for the crop; featuring easy maintenance and capacity for self-sowing, as well as staggered flowering phenologies; and finally, being attractive for pollinators and natural enemies.

The cover plants were established using an herbaceous mixture consisting of *Borago officinalis* L. (10%), *Calendula officinalis* L. (22.5%), *Coriandrum sativus* L. (10%), *Diplotaxis catholica* (L.) DC. (5%), *Echium vulgare* L. (5%), *Melilotus officinalis* (L.) Pall. (12.5%), *Nigella damascena* (L.) (5%), *Salvia verbenaca* L. (10%), *Silene vulgaris* (Moench) Garcke (10%), and *Vicia sativa* L. (10%). This mixture was sown by an electric drill with air distribution after the soil preparation by flail mower and subsequent covering of the seed with a drag. The sowing dose used was 15 Kg/Ha. The cover plants were mowed in autumn and then left to regrow.

### 2.3. Experimental Design and Sampling

To investigate the dynamics of effects of cover plants on insect biodiversity, the experiment was conducted for 3 years (2013–2015). From each farm, a field was selected, and this field was divided into two zones separated by 500–600 m. In zone 1 (*with*) three lines of cover plants were planted, while zone 2 (*without*) was kept without cover plants or weeds following the farming practices (application of residual herbicide at the beginning of the season and mechanical maintenance of weeds during the season to maintain the field without spontaneous vegetation).

The insect abundance was assessed visually and by sweeping net (observed and captured specimens were merged to perform the corresponding analyses). The observations were done by moving in a zigzag along fixed transects of 50 × 2 m during 15 min per line and 4 times per day to avoid the light and temperature gradient and obtain a more representative sample. The samples were carried out four times per year based on the phenological state of the crop (leaf growth, flowering, veraison, and harvest).

The collected specimens were preserved in cyanide to keep them intact and to avoid discoloration. All the specimens were identified to genus level using appropriate entomological literature (see [36,37,38,39,40,41,42,43,44,45,46,47,48,49,50]). Specimens are deposited in the entomological collection of the National Museum of Natural Sciences (Madrid, Spain; MNCN).

### 2.4. Data Analysis

An approach based on using statistical data analysis was developed to study the total number of insects as a measure of their biodiversity. To do that, we initially perform an exploratory data analysis of the counts of insects, comparing them among zones and between years to study their behaviour, identify abnormal situations, and detect patterns in the data. This analysis incorporates a plot of means of the number of insects between farms, zones, and years. We also build a scatterplot matrix of the counts across years to evaluate their structure of temporal autocorrelation. We then fit a generalised linear mixed model (GLMM) for count data. Mixed models allow including explicitly as sources of variability spatial and temporal factors [51] that affect the dynamics of the biodiversity. The model is specified as follows
(1)$$g\left(n_{ijkl}\right)=\mu +\alpha_{i}+\beta_{j}+{\left(\alpha \beta \right)}_{ij}+\gamma_{k}+\delta_{l}+\varepsilon_{ijkl}\{\begin{array}{c} i=1,\;2\\ j=1,\;2,\;3\\ k=1,\;2,\;3,\;4,\;5\\ l=1,\;2,\;\cdots ,\;142 \end{array}\;$$
where $n_{ijkl}$ represents the count of insects in ith zone, jth year, kth farm, and lth genus, and g is a monotonous function that linearises the relationship between the response variable and the systematic component of the model. Under the scope of count data, it is common to assume the $n_{ijkl}$ follows a Poisson distribution when its mean and variance are equal or a negative binomial when its variance is greater than its mean (overdispersion) [52]. Here, $\alpha_{i}$ is the *zone* (*fixed effect*), $\beta_{j}$ is the year (*fixed effect*), ${\left(\alpha \beta \right)}_{ij}$ is the interaction between zone and year (*fixed effect*), $\gamma_{k}$ is the farm (*random effect*), and $\delta_{l}$ is the genus (*random effect*). The farm effect ($\gamma_{k})$ accounts for the spatial variability and the grape variety. The genus effect ($\delta_{l})\;$represents the unfixed number of species per genera and helps to reduce the common overdispersion in count data due to a significant amount of zero counts. The parameters of the model in Equation (1) were estimated via maximum likelihood and assuming two possible distributions for the response variable. The fitted models are compared to choose the best probability distribution of the response variable by using a likelihood ratio (LR) contrast [53]. Statistical data analysis is performed by using R statistical software [54].

## 3. Results

### 3.1. Diversity of Insects

During the three-year research programme, a total of 7950 insects belonging to Coleoptera (1583), Diptera (1440), Hymenoptera (2291), Lepidoptera (2300), and Neuroptera (336) were collected. Of them, 3308 individuals (125 genera) were captured from the area without cover plants and 4642 (139 genera) from the area with. All the farms showed a greater number of genera and individuals in the areas with cover plants than without (Table 2). Similar results were observed when analysing the abundance of genera in order in each of the sampling areas and treatment zone (Figure 1). The complete list of genera found in each locality, year, and study zone (with and without cover plants) is provided in Appendix A.

The percentages of insect species when comparing the zone with cover plants and without always were greater in the zone with plants, being the average increase of 12.10%, having the lowest value in Carpio by only 4.16% and the greatest increase in Villamanta (18.82%). Similar results were obtained when considering individuals, with the mean value of 40.01%, obtaining again the lowest value in Carpio (8.66%) and the highest in Sardón (71.28%).

The analysis of genera identified in areas with and without cover plants showed that areas with plants had a higher number of genus than the area without (Table 2). The five most frequent genus in all farms were *Apis*, *Chrysoperla*, *Coccinella, Eristalis*, and *Andrena* (Table 3). All these genera were also the most abundant in areas with cover plants, where they represented 27.18% of the total individuals. In areas without plants, the first four genera previously listed were also the most captured, accompanied by *Sphaerophoria*, representing a total of 24.12%. The other genus (134 in zone 1 and 120 in zone 2) had a very low frequency of abundance.

The biology of these genera shows two clear key functions within the agricultural ecosystem. *Apis* and *Andrena* carry out functions of pollination, while *Coccinella* and *Chrysoperla* are predators of aphids whose behavior regulates aphid populations. In addition, genera such as *Eristalis* and *Sphaerophoria* perform both functions (pollination and predation).

### 3.2. GLMM Modelling

An exploratory data analysis was initially performed to describe the behaviour of the total number of insects under the assessed determinants. Figure 2 shows the changes in the average number of insects between zones, farms, and years. There is a clear trend of the average number of insects increasing through the years in all farms. However, this trend does not seem to be the same between farms, which differs the rate of change. In most cases, the zones with cover plants have a higher number of insects in comparison with the zones without cover plants. Additionally, there is an interaction effect among the zones and the years.

Additionally, a study of the structure of temporal correlation between the count of insects through the years was conducted. To do that, we plotted a scatterplot matrix among the total number of insects in each observed year and tested the statistical significance of Pearson’s correlation coefficient of the counts among years. Figure 3 presents the results of the analysis and indicates that the number of insects is significantly correlated between the years. The funnel-shaped clouds of points also reveal an effect of unequal variability of the number of insects between years that talks about the complexity of insect population dynamics among the contrasting farming environments.

We fitted two-count GLMMs based on Equation (1) by considering a Poisson and a negative binomial response. Table 4 presents the statistics for the goodness of fit to the estimated models in each case. The LR test shows that the model has a better fit using a negative binomial distribution for the response variable since it rejects the null hypothesis that the ratio is less than 1 in the case of the Poisson distribution, which means that the variance of the count of insects increases more rapidly than their mean and the negative binomial distribution is more accurate as a probabilistic schema for the total number of insects. Moreover, the other statistics of the goodness of fits, such as AIC and BIC, are considerably lower for the model that assumes the negative binomial distribution for the response variable. And, finally, we concluded that the preferred model is suitable to explain the total number of insects as a function of the examined systematic component (i.e., zones, years, farms, and genera) since the deviance statistic is also statistically significant.

Table 5 presents the Analysis of Deviance, and Table 6 presents the summary of the estimation for the selected negative binomial GLMM, respectively. The results show that the parameters related to fixed effects are all statistically significant. This latter means: (i) there is an interaction effect between the zone and the year that implies that the main effect of the zone is not constant through the years, and their signs are positive, which tells that increase in the average of the total number of insects in both zones through time; (ii) there are also differences due to both main effects, an increase of the number of insects with time and a smaller number of insects in zones without cover plants.

We finally performed hypotheses testing over the fixed and random effects to understand the behaviour of the total number of insects under the examined factors. For the fixed effects (zones and years), Tukey multiple comparisons strategy was used to compare the differences among the zones without and with cover plants through the years. Thus, for each year, we tested the null hypothesis that there is not a difference in the mean of the number of insects between the zone without cover plants and the zone with cover plants. The results are shown in Figure 4a and indicate that in the years 2013 and 2015, there was a statistically significant difference, where the zone with cover plants had more insects than the zone without cover plants on average.

Regarding the random effect associated with the farms, the estimated parameters, related to the random intercepts (see Figure 4b), show that Carpio and Sardón were statistically significant contrasting environments. Sardón had a higher total number of insects while Carpio had a lower count of insects. The other three farms did not show significant differences since the confidence intervals for the random intercepts contain the value zero.

## 4. Discussion

The importance of the use of cover plants to increase the diversity of plant species that directly or indirectly promote the diversity of invertebrates is well known [55,56]. Moreover, an increased mixture of plant species richness can potentially support increasing numbers of specialised feeders [55]. According to our first working hypothesis, this significant association between plants and insects can increase the number of species and individuals over time. Several studies have remarked the influence that cover plants have on the abundance of insects and their role in pest management [57,58,59,60,61]. The results of the present work show a similar pattern in the attraction of insects because in all farms, more species and individuals were observed in areas with cover plants than without them. However, the improvement obtained showed clear differences between Carpio and Sardón. These differences lie primarily in their location and the environmental conditions of the surrounding countryside. In the case of Carpio, the field is located in an arid zone without a nearby natural environment, while Sardón is near the stream of a river in an area of fluvial vegetation. This shows that the habitat surrounding the fields may play a very important role in the rate and speed of biodiversity enhancement. However, as noted in the current study, it is not a limiting factor.

According to the second hypothesis, cover plants may contribute to the conservation of insects and biodiversity enhancement. The increase in vegetation structural complexity and diversity in vineyards is important to increased resources (i.e., pollen and nectar) or refuge in the field [62,63,64]. Numerous researchers have studied the effect of the management of cover plants in agricultural environments. However, these studies were based only on one growing season [57,58,59,60,61,65], unlike our study that analyzes the evolution of insect communities over three years. Looking only at the data obtained during the first year, we see that the results are very similar to previous studies, and a significant increase in the number of insects was observed. The analysis of three years data set showed a clear trend in the increase of the number of insects over the years in all the farms, both in the areas with plants and without plants. This increase in both areas may be due to the establishment of the durable cover plant over time, which plays a fundamental role in the creation of refuge areas and in the conservation of insects in the agricultural ecosystem. The only study we have been able to find that covers more than one crop season was carried out in an alfalfa field where a wildflower margin was used for three years [49]. The results of the alfalfa field also showed an increase in the number of insect species (102.47%) and individuals (97.64%) at the end of the experiment. These numbers are far from those observed in vineyards (12.10% species and 40.01% individuals), presumably because the difference in the biology and flowering of crops, such as alfalfa flowers, attracts numerous pollinators and beneficials, while those of the vineyard do not.

Furthermore, the enhancement of insect biodiversity due to cover plants may play an important role as a natural biological control. Therefore, its use is also recommended as an Integrated Pest Management (IPM) tool to balance and eventually reduces the long-term pest populations increasing the natural enemies’ community [62,66,67,68,69]. Our results showed that within the most abundant genera were *Coccinella* and *Chrysoperla,* who are predators of aphids, and *Sphaerophoria,* who perform both pollination and predation functions. These observations coincide with those obtained by other authors [57,59,60,70,71] and prove that cover plants are a great tool for IPM.

## 5. Conclusions

The implementation of cover plants in agricultural landscapes results in biodiversity enhancements through the attraction of pollinators and beneficial insects. These effects are mainly because cover plants create a perfect reservoir area with refuge and food (pollen, nectar, alternative prey) that allow them to maintain and extend their populations. In addition, we concluded that the implementation of permanent plant cover inter-row crops could be an important biological conservation strategy.

## Figures and Tables

**Figure 1 insects-12-00740-f001:**
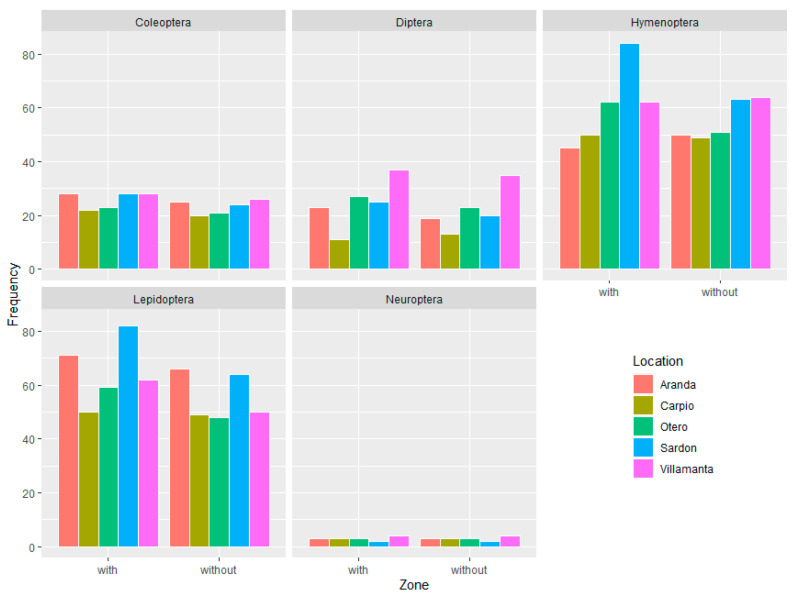
The abundance of genera by locality and treatment zone by insect order.

**Figure 2 insects-12-00740-f002:**
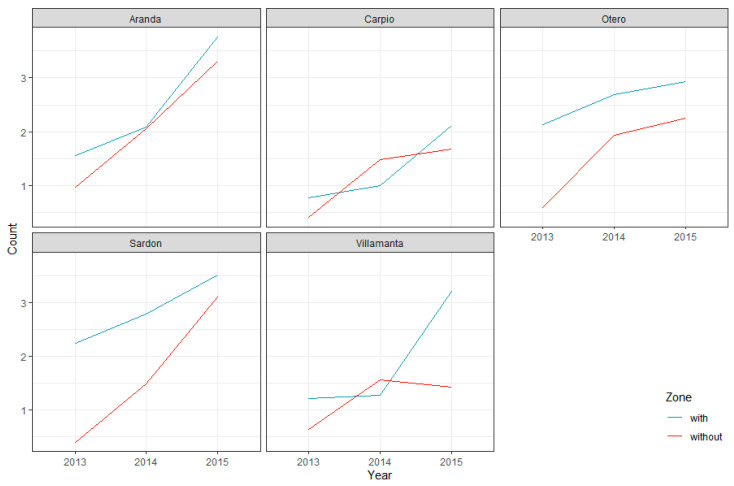
Plot of the means of the total number of insects between zones across the farms through the years.

**Figure 3 insects-12-00740-f003:**
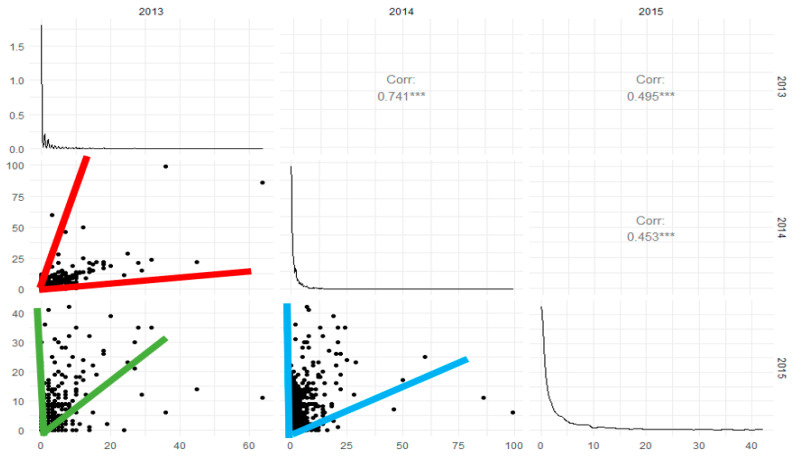
Scatterplot matrix of the total number of insects through the years.

**Figure 4 insects-12-00740-f004:**
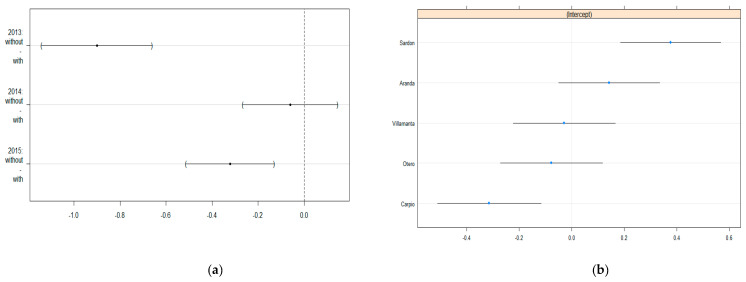
Ninety-five percent confidence intervals for the fixed and random effects of the fitted negative binomial GLMM; (**a**) 95% confidence intervals for Tukey multiple comparisons between zones through years; (**b**) 95% confidence intervals for the random intercepts of the farms.

**Table 1 insects-12-00740-t001:** Farm location and GPS information.

Province	Place	GPS Coordinates	Grape Variety	Field Area (ha)
Burgos	Aranda de Duero	41°37′22.51″ N, 3°41′17.44″ W	Tempranillo	5
Madrid	Villamanta	40°18′00.61″ N, 4°07′17.83″ W	Garnacha	4
Toledo	El Carpio de Tajo	39°49′34.70″ N, 4°26′38.80″ W	Tempranillo	5
Toledo	Otero	39°59′36.00″ N, 4°31′42.60″ W	Tempranillo	4
Valladolid	Sardón de Duero	41°37′01.00″ N, 4°24′39.00″ W	Tempranillo	4.5

**Table 2 insects-12-00740-t002:** The abundance of genera and individuals by vineyard farm and treatment.

	Aranda de Duero		El Carpio de Tajo		Otero		Sardón de Duero		Villamanta	
	with Cover	without Cover	with Cover	without Cover	with Cover	without Cover	with Cover	without Cover	with Cover	without Cover
Genera	86	78	75	72	82	74	106	91	101	85
Individuals	1053	899	552	508	1100	679	1211	707	726	515

**Table 3 insects-12-00740-t003:** Top 5 most abundant genera in areas with and without cover plants.

With Cover Plants			Without Cover Plants		
Genus	Individuals	%	Genus	Individuals	%
*Coccinella*	301	6.35	*Coccinella*	235	7.10
*Apis*	285	6.01	*Eristalis*	153	4.63
*Eristalis*	268	5.65	*Sphaerophoria*	147	4.44
*Andrena*	223	4.70	*Apis*	141	4.26
*Chrysoperla*	212	4.47	*Chrysoperla*	122	3.69
Total	1289	27.18		798	24.12

**Table 4 insects-12-00740-t004:** Statistics of goodness of fit for fitted count regression models.

Test	Poisson	Negative Binomial
Likelihood ratio (LR)	2.29	0.75 ***
Deviance (D)	15,453.7 ***	11,598.6 ***
AIC	15,469.7	11,616.6
BIC	15,520.5	11,673.8

*** [0, 0.001]; ** [0.001, 0.01]; * [0.01, 0.05]; ^∙^ [0.05, 0.1]; [0.1, 1].

**Table 5 insects-12-00740-t005:** Analysis of Deviance Table (Type II Wald chi-square tests) in the fitted negative binomial regression model for the total number of insects.

Source	Chisq	Df	*p*-Value	
Zone	56.04	1	7.117 × 10^−14^	***
Year	412.57	1	<2.2 × 10^−16^	***
Zone:Year	42	2	7.553 × 10^−10^	***

*** [0, 0.001]; ** [0.001, 0.01]; * [0.01, 0.05]; ^∙^ [0.05, 0.1]; [0.1, 1].

**Table 6 insects-12-00740-t006:** The estimated regression coefficients in the fitted negative binomial regression model for the total number of insects.

**Random Effects**			
**Groups**	**Name**	**Variance**	**Std. Dev.**
Genus	(Intercept)	2.40	1.55
Farm	(Intercept)	0.06	0.25
**Fixed Effects**			
**Parameter**	**Estimate**	**Std. Error**	
(Intercept)	−0.78	0.19	***
Zone:without	−0.90	0.10	***
Year:2014	0.32	0.09	***
Year:2015	1.07	0.08	***
Zone:without-Year: 2014	0.84	0.13	***
Zone:without-Year: 2015	0.58	0.13	***

*** [0, 0.001]; ** [0.001, 0.01]; * [0.01, 0.05]; ^∙^ [0.05, 0.1]; [0.1, 1].

## Data Availability

The data presented in this study are available from the corresponding author, upon reasonable request.

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
