# Peer review of "Enhancement of the Diversity of Pollinators and Beneficial Insects in Intensively Managed Vineyards"

_insects, 2021, doi:10.3390/insects12080740_

Round 1

Reviewer 1 Report

Enhancement of the diversity of pollinators and beneficial insects in intensively managed 
vineyards, by Peris-Felipo et al.
This study compared the abundance and species richness of insects in five matched vineyard plots, 
which were either sown with herbaceous vegetation between the vine rows or left untreated. Insect 
abundance increased in all plots during the three years of the study, and was higher in the coverplant plots than in the controls. Since the most abundant insect genera sampled were pollinators 
and predators, the authors conclude that the added vegetation likely contributes to pollination and 
pest control in the vineyards. 
Habitat manipulations to conserve local beneficial insects are widely practiced in recent years, but 
the results are inconsistent among crops and sites (1, 2).
1. Karp, D. S., Chaplin-Kramer, R., Meehan, T. D., Martin, E. A., DeClerck, F., Grab, H., ... & 
Wickens, J. B. (2018). Crop pests and predators exhibit inconsistent responses to surrounding 
landscape composition. Proceedings of the National Academy of Sciences, 115(33), E7863-E7870
2. Dainese, M., Martin, E. A., Aizen, M. A., Albrecht, M., Bartomeus, I., Bommarco, R., ... & 
Steffan-Dewenter, I. (2019). A global synthesis reveals biodiversity-mediated benefits for crop 
production. Science advances, 5(10), eaax0121.
This justifies studies that test the effects of added vegetation for a specific cropping system and 
geographical areas, as was done here. While the study question is not novel, insights can potentially 
be gained for local agricultural management in Spanish vineyards. Unfortunately, the experiment 
reported here is not described in sufficient detail to determine the validity of the sampling protocol; 
The only dependent variable that was statistically analyzed is the total number of insects; and the 
sampling methods (netting and visual search) probably over-represent pollinators and large 
predators, while pests may be under-represented. 
Title: 
“pollinators” are a subgroup of “beneficial insects”. Perhaps “pollinators and predators”?
Introduction: 
The authors should acknowledge the many studies on plant diversification in agricultural crops 
(1,2) and in vineyards in particular (3, 4 and references therein). The rational for doing yet another 
similar study needs to be better explained.
(3) Paiola, A., Assandri, G., Brambilla, M., Zottini, M., Pedrini, P., & Nascimbene, J. (2020). 
Exploring the potential of vineyards for biodiversity conservation and delivery of biodiversitymediated ecosystem services: A global-scale systematic review. Science of the Total 
Environment, 706, 135839.(4) Möller, G.; Keasar, T.; Shapira, I.; Möller, D.; Ferrante, M.; Segoli, M. Effect of Weed 
Management on the Parasitoid Community in Mediterranean Vineyards. Biology 2021, 10, 7. 
https://dx.doi.org/10.3390/biology10010007
The hypotheses that are mentioned in the Discussion (l. 225, 238) need to be introduced and
developed in the Introduction section.
Methods:
Table 1: What were the sizes of the plots? The grape varieties?
l. 78-79: “…and finally, being attractive for pollinators and natural enemies 79 but not for crop 
pests”. How was the suitability of the cover plants for beneficial and pest insects determined?
l. 91-92: What was the distance between zone 1 and zone 2? Were they separated by a buffer 
zone that was not sampled?
l. 93: What (if any) was the non-crop vegetation in zone 2? What was its vegetation cover? Were 
insects sampled only from the non-crop vegetation, or from the grape vines as well?
l. 95-96: What were the months/seasons of sampling? What was the duration of sampling per 
transect? Was sampling duration (i.e. sampling effort) constant across seasons, sites and years?
Equation 1: Why is genus included in the model as an explanatory variable?
Results:
Table 2: Were there statistically significant differences in the number of individuals and taxa 
between matched plots with and without cover vegetation?
Table 3: Was there a significant difference in insect community composition (considering all 
species, or just the dominant ones) between the two zones? 
l. 165-167: The correlation between 2013-2015 was stronger than between 2014-2015. This 
seems to contradict the statement that “there is a stronger correlation when the assessments are 
closer in the time and weaker when the temporal lag increases “. 
Fig. 2: This figure can be deleted, and the correlation coefficients can be reported in the text.
l. 152-175: These results are not part of the GLMM and should come under a separated heading.
l. 167: I was not able to see the funnel shaped clouds.
Table 4: Why not test directly whether the data conforms to the assumptions of the 
poisson/negative binomial distributions? What was the rationale of pooling the four samples taken 
within each year, rather than also testing for the effect of season?
Table 5: It would be helpful to run R’s anova function on the GLMM model to evaluate the overall 
effects of zone, year and their interaction on insect abundance. The present table reports comparisons between pairs of levels within each explanatory variable, as well as interactions 
between pairs of levels.
Discussion, simple summary and abstract:
Netting mainly captures large flying insects, while visual search often targets mobile and large 
plant-dwellers (such as beetles and spiders). Smaller mobile insects (such as whiteflies) and sessile 
plant-feeder (such as aphids) are likely to be under-represented. This sampling bias might 
contribute to the conclusion that bees, flies and coccinellids are the dominant insect in the 
vineyards. A short discussion of the biases generated by the sampling method is warranted.
The effects of the cover plant on pollination and pest control were not measured. The discussion 
of such potential effects is highly speculative and should be reduced.
l. 248-250: This sentence implies that the planted and control zones were not sufficiently far apart, 
so that insects spilled over from the seeded to the unseeded zone. In addition, it suggests that noncrop vegetation developed over the three years of the study in the control zones as well (perhaps 
spreading from the seeded zones), i.e. that vegetation cover and composition varied over time in 
the control plots. Both of these potential issues might compromise the experimental design, and 
should be explicitly addressed.
Specific comments:
l. 77: Please define ‘allochthone’. Do you mean non-native plant species?
l. 85: “planted” – should be “sown”?
l. 138-139: should be “genera”
Table 3: Please provide the family and order of the most abundant insects found.
l. 269: Should be “beneficial

Author Response

Dear Reviewer,

Thanks for your suggestions and comments.

Please find below our comments.

Hyphotesis

We have added hypothesis in the introduction and in the summary.

Please find in between lines 64-70.

Methods: What were plot size? Grape varieties? Cove plan management? Sampling information.

In Table 1 we have added information on plot size and grape varieties. Moreover, we added information about the cover plant management. Finally, we have better detailed sampling protocol.

Please find this information between lines 94 and 101.

Equation 1: Why is genus included in the model as an explanatory variable?

A/:

Due to the type of experiment is not possible neither having a fixed number of genera location by location or year by year, nor a fixed number of species for each genus. This kind of unbalanced must be accounted for in models and predictions because might increase the overdispersion effect that is common in count data and affect the validity and scope of the statistical conclusions. Thus, to avoid that situation, we decided to include the genus as a random factor in the GLMM.

We have included in the lines 123 to 125 the following statement:

The farm effect ( account for the spatial variability and the grape variety. The genus effect ( represents the unfixed number of species per genera and helps to reduce the common overdispersion in count data due to a significant amount of zero counts.”

Table 2: Were there statistically significant differences in the number of individuals and taxa

between matched plots with and without cover vegetation?

And     

Table 3: Was there a significant difference in insect community composition (considering all

species, or just the dominant ones) between the two zones?  

Both tables show descriptive statistics related to the abundance of genera and individuals and the most abundant genera collected in both zones.

  1. 152-175: These results are not part of the GLMM and should come under a separated heading.

A/:

In fact, these results are a constitutive part of the strategy of modelling. The exploratory data analysis (EDA) of the fixed effects (zone and year) of the proposed model helps us and the reader to understand the behaviour of the data beyond the equations, identifies abnormal situations or patterns in the data that were unknown or not initially included. These exploratory data tools are common in regression and experimental design modelling. Thus, we have decided to do not include these results under a separated heading.

  1. 165-167: The correlation between 2013-2015 was stronger than between 2014-2015. This 
    seems to contradict the statement that “there is a stronger correlation when the assessments are 
    closer in the time and weaker when the temporal lag increases “. 

A/:

We have removed this sentence from the manuscript.

Fig. 2: This figure can be deleted, and the correlation coefficients can be reported in the text and l. 167: I was not able to see the funnel shaped clouds.

A/:

We have decided to do not remove the figure since the scatterplots show the overdispersion effect of the phenomena. To help the reader, we have highlighted the funnel shapes in the Figure. 

Table 4: Why not test directly whether the data conforms to the assumptions of the 
poisson/negative binomial distributions? What was the rationale of pooling the four samples taken 
within each year, rather than also testing for the effect of season?

A/:

The approach of testing directly over the observed sample if the counts come from a Poisson or negative binomial distribution is an incorrect one since in that case we would be assuming that the underlying mechanism that generates the data is the same for all of them, e.g., in the case of a Poisson distribution that its mean is the same. This would reduce the analysis to the univariate statistics. The idea of fitting a GLMM is account for several sources of variability that can explain the phenomenon. Thus, as in all regression models, the distribution assumptions are assessed under the estimated model and its residuals. We have followed the good statistical practices.

Table 5: It would be helpful to run R’s anova function on the GLMM model to evaluate the overall 
effects of zone, year and their interaction on insect abundance. The present table reports comparisons between pairs of levels within each explanatory variable, as well as interactions 
between pairs of levels.

A/:

We have additionally included the analysis of deviance table.

I 248-250: experimental design should be addressed.

We included this information in the M&M section.

Reviewer 2 Report

The paper discusses how to increase the diversity of pollinators and beneficial insects in intensively managed vineyards. The topic is interesting and current because, although the wine agro-ecosystem management in Europe is extremely varied in terms of sustainability, in most cases it is not very attractive and may be even inhospitable for pollinators and beneficial insects.
Sowing cover plants between rows can certainly be one of the useful ways to increase diversity.
The experimental design appears well set, however my main criticism is the authors are stingy with details in some important aspects (see comments) both for methods and for results. Precisely for the double reason that the vineyard system is extremely varied and that the sowing of cover plants is only one of many possible interventions to increase sustainability, for this type of intervention to be useful, it should be specified in many details.

70-72 All the established agricultural practices ....... remained unchanged
Please give more details. What kind of winter tillage, what kind of tillage during the growing season? During the growing season, among the established practices, were mowings carried out to eliminate the spontaneous flora? How many? none, 1, 2, 3 ...? If the mowings were carried out among the established practices, how can these be reconciled with the sowing of cover plants?

91-92 In the zone 1 ... altering the conditions
Please give more details. a) on the choice of fields. Were the fields of similar size among the 5 farms? b) about cover plants sowing methods: Was sowing preceded by plowing? Was a sod seeding type instead?

98 All the specimens were identified to species level using appropriate entomological literature (see [36])
The cited reference does not appear sufficient to identify 7950 insects at the species level

128 In a text that includes both plants and insects, the term "species" should never be used alone, but always accompanied: "plant species" or "insect species"

136, 143, 148, 262 Sphaerophoria

136 Top most abundant genera together totaled 27.18% and 23.12%. What about the remaining 70% and more? What about the genus Lasioglossum, usually the most abundant among bees (not including Apis) in any sampling of flower visitors?

217 Significance appears not indicated

224 An increased plant species richness ... Was a list of plant species occurring in the lines made before sowing cover plants? Since details on the sowing of cover plants are missing (see comment 91-92) it is not clear whether the "increased plant species richness" is obtained because the number of species in the cover plant mix is ​​greater than the number of pre-existing species, or if it is due to the sum of the new species sown and a part of the pre-existing ones that have remained.

225 According to our first working hypothesis and 238 According to the second hypothesis
Two working hypotheses are stated here. The linearity of the reasoning that is pursued here is good. However, neither in the abstract, nor in the introduction nor in the methods, these two working hypotheses are stated with equal clarity. A better link between the research aim (line 32), methods (line 89), results and discussion would be needed

Author Response

Dear Reviewer,

Thanks for your suggestions and comments.

Please find below our comments.

70-72 and 91-92 - Management details and plot size.

In Table 1 we have added information on plot size and grape varieties. Moreover, we added information about the cover plant management. Finally, we have better detailed sampling protocol.

Please find this information between lines 94 and 101.

98 – references to insect identification.

We added all references related to the identification. Previously we just mentioned one because in this book are mentioned all the added ones.

Please find this information in line 106

128 – species concept

We check the MS and added insect or plant depending the case.

136, 143, 148, 262 Sphaerophoria.

Thanks for the observation. It was a mistake.

136 – Top most abundant species.

Other ones showed smaller values. The case of Lasioglossum, we found only 20 specimens during all the study.

225 – Hypothesis

We have added hypothesis in the introduction and in the summary.

Please find in between lines 64-70.

Reviewer 3 Report

The work of Peris-Felipo and collaborators evaluates the effect of using cover plants between vineyards lines in the insect diversity. The study was carried out in five intensively managed vineyards from Spain during three consecutive years. The design of the study is robust and well though.  The authors selected focal plant species known to attract different pollinators and natural enemies but not crop pests and recorded insect diversity among vineyards with vs without flowering plants covering between vineyards lines. They showed that incorporating covering plants in the vinyards significantly increase the diversity of insects. These findings have a clear impact in future guidelines for vineyards management. An increase in the diversity of insect pollinators and natural enemies (predators and parasitoids) is key for the sustainable pest control provided by insects.

However, I consider that several changes are needed in order to improve the manuscript. I outlined my suggestions below:

  • Goals, hypothesis and predictions expected to achieve in the work are not clearly stated. They are a key part of the work and need to be mentioned in the introduction, simple summary and abstract sections. In the discussion authors mentioned about two working hypothesis that I considered are not clearly state in the introduction.
  • I would mention which are the pollinators (from literature) of the selected flowering plants. Are generalist plant species or are pollinated by specific groups of pollinators (i.e. Apis mellifera, other bees, diptera, butterflies)? (Lines 81-87 in material and methods).
  • Sampling effort needs to be detailed. Authors mentioned “four times per year”, how many days? Hours? Sampling effort was the same in the five vineyards? Rarefaction curves were calculated? (Lines 93-95). Observations were undertaken only during daytime? There are many pollinator species such as moths that only flights during nighttime.
  • One of my main concerns is about the selection of groups to compare in the data analyses and the results shown. Authors mentioned that insects were identified to species level (line 98) but they only show abundance of genera and individuals in the analyses (Table 2). I think it is important to show as supplementary information the complete list of species captured indicating whether is know from the literature if they are pollinators or beneficial insects from the IPM point of view (predators, parasitoids, etc.). The title of the work suggest that diversity of pollinators and beneficial insects will be shown but the information is fragmentary, and only five genera (from more than 80 in averaged recorded) are mentioned in Table 3. Findings with regard to number of species (Fig. 1 and 2) are OK, my main concern is about the genera classification.
  • I think it is necessary to show a comparison of the proportion of different pollinators groups (diptera, hymenoptera, butterfly, etc.) and beneficial insects (a possible classification could be predator and parasitoids) among the five vineyards (e.g. graph bars). I consider this information is very important and is not shown in the work.  I suggest comparing number of species or morphospecies present in each class instead of genera. I suggest using a classification using functional groups instead of genera (sensu Fenster et al. 2004 for pollinators and maybe in predators and parasitoids for beneficial insects), it depends on the diversity of insects that authors have captured (information not shown). I I think that papers from Reverté et al. (2014, 2016) could be useful in determining the pollinator classification since they work in the same geographic region.
  • Pollinators groups are not equally important and efficient as pollinators. Depending the results obtained analyzing data as I suggested before, these new results will allow the authors to discuss about it.  

Minor observations:

Line 125: …from the area with cover plants.

References:

Fenster, C. B., Armbruster, W. S., Wilson, P., Dudash, M. R., & Thomson, J. D. (2004). Pollination syndromes and floral specialization. Annu. Rev. Ecol. Evol. Syst., 35, 375-403.

Reverté Saiz, S., & Retana Alumbreros, J. (2014). Relationship between floral colour and pollinator composition in four plant communities.

Reverté, S., Retana, J., Gómez, J. M., & Bosch, J. (2016). Pollinators show flower colour preferences but flowers with similar colours do not attract similar pollinators. Annals of botany118(2), 249-257.

Author Response

Dear Reviewer,

Thanks for your suggestions and comments.

Please find below our comments.

Goals and Hypothesis

We have added hypothesis in the introduction and in the summary.

Please find in between lines 64-70.

Methods: What were plot size? Grape varieties? Cove plan management? Sampling information.

In Table 1 we have added information on plot size and grape varieties. Moreover, we added information about the cover plant management. Finally, we have better detailed sampling protocol.

Please find this information between lines 94 and 101.

Why we consider genus?

Due to the type of experiment is not possible neither having a fixed number of genera location by location or year by year, nor a fixed number of species for each genus. This kind of unbalanced must be accounted for in models and predictions because might increase the overdispersion effect that is common in count data and affect the validity and scope of the statistical conclusions. Thus, to avoid that situation, we decided to include the genus as a random factor in the GLMM.

We have included in the lines 123 to 125 the following statement:

The farm effect ( account for the spatial variability and the grape variety. The genus effect ( represents the unfixed number of species per genera and helps to reduce the common overdispersion in count data due to a significant amount of zero counts.”

I think it is necessary to show a comparison of the proportion of different groups.

We add a comparison between different orders by locality and treatment in lines 137-138 and Figure 1.

Round 2

Author Response

Dear reviewer,

We appreciate your evaluation and your points were considered in both manuscript reviews. Howeversome of your comments are outside the scope of the work carried out and the objectives proposed which could be analyzed as part of future work.

Kind regards

Reviewer 3 Report

All my suggestions have been incorporated in the manuscript. The only comment is that I consider that the hypothesis of the work can be more concisely stated in the introduction, there is a previous explanation that can be shortened or omitted. Please see the suggestions provided by:

Farji-Brener, A. G. (2003). Uso correcto, parcial e incorrecto de los términos" hipótesis" y" predicciones" en ecología. Ecología Austral13(02), 223-227.

https://bibliotecadigital.exactas.uba.ar/download/ecologiaaustral/ecologiaaustral_v013_n02_p223.pdf

Author Response

Dear Reviewer,

Thank you for your comments. We have worked on the hypothesis following the document you suggested.

Kind regards
